# Adult-Acquired Flatfoot Deformity: Combined Talonavicular Arthrodesis and Calcaneal Displacement Osteotomy versus Double Arthrodesis

**DOI:** 10.3390/jcm11030840

**Published:** 2022-02-05

**Authors:** Sebastian Fischer, Julia Oepping, Jan Altmeppen, Yves Gramlich, Oliver Neun, Sebastian Manegold, Reinhard Hoffmann

**Affiliations:** 1Department of Foot and Ankle Surgery Berufsgenossenschaftliche Unfallklinik Frankfurt am Main, 60389 Frankfurt, Germany; Julia.oepping@gmx.de (J.O.); Oliver.neun@bgu-frankfurt.de (O.N.); sebastian.manegold@bgu-frankfurt.de (S.M.); 2Department for Trauma and Orthopaedic Surgery Berufsgenossenschaftliche Unfallklinik Frankfurt am Main, 60389 Frankfurt, Germany; Jan-niklas.altmeppen@bgu-frankfurt.de (J.A.); Yves.gramlich@bgu-frankfurt.de (Y.G.); Reinhard.hoffmann@bgu-frankfurt.de (R.H.)

**Keywords:** adult-acquired flatfoot deformity (AAFD), double arthrodesis, talonavicular arthrodesis, calcaneal displacement osteotomy, posterior tibial tendon dysfunction (PTTD)

## Abstract

Background: Adult-acquired flatfoot deformity due to posterior tibial tendon dysfunction (PTTD) is one of the most common foot deformities among adults. Hypothesis: Our study aimed to confirm that the combined procedures of calcaneal displacement osteotomy and talonavicular arthrodesis are equivalent to double arthrodesis. Methods: Between 2016 and 2020, 41 patients (13 male and 28 females, mean age of 63 years) were retrospectively enrolled in the comparative study. All deformities were classified into Stages II and III of PTTD, according to Johnson and Strom. All patients underwent isolated bony realignment of the deformity: group A (*n* = 19) underwent calcaneal displacement osteotomy and talonavicular arthrodesis, and group B (*n* = 23) underwent double arthrodesis. Measurements from the Foot Function Index-D (FFI-D) and the SF-12 questionnaire were collected, with a comparison of pre- and post-operative radiographs conducted. The mean follow-up period for patients was 3.4 years. Results: The mean FFI-D was 33.9 (group A: 34.5; group B: 33.5), the mean SF-12 physical component summary was 43.13 (group A: 40.9; group B: 44.9), and the mean SF-12 mental component summary was 43.13 (group A: 40.9; group B: 44.9). The clinical data and corrected angles showed no significant intergroup differences. Conclusion: Based on the available data, our study confirmed that the combined procedures of talonavicular arthrodesis and calcaneal shift, with preservation of the subtalar joint, can be considered equivalent to the established double arthrodesis, with no significant differences in terms of clinical and radiological outcomes.

## 1. Introduction

Adult-acquired flatfoot deformity (AAFD) due to posterior tibial tendon dysfunction (PTTD) is one of the most common foot deformities among adults. The prevalence of AAFD is as much as 10% in those aged >40 years, and it predominantly affects women [1,2]. In addition to flattening of the medial longitudinal arch, patients present a weakness of the bipedal tiptoe position, with the unipedal tiptoe position being completely impossible in most cases, and pain projection is predominantly to the degenerated tibialis posterior tendon near the insertion [3,4]. The insufficiency of the degenerated posterior tibial tendon further causes an aggravating valgus tilt of the hindfoot with a positive “too many toes sign”.

The staging of PTTD, according to Johnson and Strom (Stages I–III), along with its modification according to Raikin and extension according to Myerson (Stage IV) has become established [5,6,7]. Failure of conservative therapy using physiotherapy, orthotic insole, and shoe fitting is an indication for surgical intervention. For mild deformities, especially in younger adults, joint-preserving osteotomies and soft-tissue balancing reconstructive procedures are used [8,9]. Stage III, according to Johnson and Strom, describes a longitudinal or transverse rupture of the posterior tibial tendon on magnetic resonance imaging (MRI) with fixed deformity in addition to the above symptoms [10,11]. In these cases, restoration of the physiological axes by tarsal arthrodesis has become an established treatment [12]. In most cases, double or triple arthrodesis, combined with calcaneal displacement osteotomy, are used. Depending on the intraoperative findings, this operation can be extended by a soft-tissue balancing procedure. The indication for each procedure is largely based on the surgeon’s experience and remains the subject of clinical research. However, a proportional decrease in the number of triple arthrodesis performed in favour of double arthrodesis can be observed [10,13].

In the present study, we compared two isolated bony realignments for AAFD: combined calcaneal displacement osteotomy with talonavicular arthrodesis and double arthrodesis (talonavicular arthrodesis and subtalar arthrodesis). Biomechanical studies and clinical experience reports had previously confirmed the combined procedure as an adequate treatment for AAFD [14,15]. The study was based on the hypothesis that the combined procedure of talonavicular arthrodesis and calcaneal medial displacement osteotomy, with preservation of the subtalar joint, can demonstrate patient satisfaction comparable to the established double arthrodesis, based on clinical scores.

## 2. Patients and Methods

### 2.1. Population

Between 2016 and 2020, 41 patients (13 males (31.7%), 28 females (68.3%); mean age of 63 years (range: 35–81 years)) were retrospectively enrolled in this comparative study. The demographics of both groups were comparable (Table 1). The radiological angles were also comparable (Table 2). All patients were seen in the foot surgery consultation clinic of the study center (Figure 1). The AAFD diagnoses were always made on the basis of clinical examination, obligatory weight-bearing radiographs, and, where necessary, additional MRI scans. All patients underwent isolated bony realignment of the deformity: group A (*n* = 19) underwent calcaneal displacement osteotomy and talonavicular arthrodesis, and group B (*n* = 23) underwent double arthrodesis.

The mean follow-up duration for clinical and radiographic outcomes was 3.4 (range: 1.5–5.9) years.

#### 2.1.1. Inclusion and Exclusion Criteria

*Inclusion criteria.* Patients with a minimum age of 18 years were included. There was no maximum age limit. Written informed consent was required prior to participation in this study. Only patients with AAFD due to posterior tibial tendon dysfunction Stage II and III, according to Johnson and Strom (IIb and IIIa, b according to Raikin et al.), who underwent surgery at the study center, were included [6]. In the case of more minor deformities, the authors predominantly chose soft-tissue balancing procedures or conservative treatments.

#### 2.1.2. Exclusion Criteria

Patients with AAFD due to posterior tibial tendon dysfunction Stage I, according to Johnson and Strom, and Stage IV, according to Raikin et al., were not included in the study. The same applied to patients who, in addition to bony realignment, received a soft-tissue balancing procedure such as transfer of the flexor hallucis longus or flexor digitorum longus tendon. Patients undergoing permanent pain therapy were also excluded.

### 2.2. Surgical Procedure

The decision to choose one of the compared procedures was essentially determined from the experience of the particular surgeon. No histological examination of the posterior tibial tendon was performed during surgery. The function of the posterior tibial tendon was assessed on the basis of the previous clinical examination.

*Medial calcaneal displacement osteotomy and talonavicular arthrodesis* (Group A). In the first step, a V-shaped osteotomy was performed via an oblique approach to the lateral calcaneus with medial displacement of the back portion of the calcaneus. Osteotomy was performed using either an oscillating bone saw or bone chisel. Depending on the extent of the deformity, a medial displacement of approximately 8–10 mm was considered optimal. Fixation was achieved by means of two percutaneously inserted lag screws with diameters of 4.0–6.5 mm. The second step was talonavicular arthrodesis via a dorsal approach, usually with interposition of an iliac crest bone graft. Fixation was performed using two or three screws with diameters of 4 mm or a combination of screws and Nitinol compression implants (Figure 2a,b and Figure 3a,b).

*Double Arthrodesis* (Group B). Firstly, subtalar arthrodesis was performed via a lateral subtalar approach. Care was taken to ensure thorough resection of the articular surfaces and extensive release to achieve adequate correction of the valgus deformity and avoid the development of pseudarthrosis. Here, too, the insertion of autologous cancellous bone was considered obligatory. Fixation was achieved by means of two percutaneously inserted lag screws (diameter of 6.5 mm). The subsequent talonavicular arthrodesis was performed as previously described (Figure 4a,b and Figure 5a,b).

### 2.3. Rehabilitation Protocol

The recommended rehabilitation protocol required patients to wear an orthotic boot for 10–12 weeks, with 6 weeks of sole contact in a neutral position. Removal of the boot was permitted for personal hygiene and non-weight-bearing exercise. A radiograph was taken after 6 weeks. Accompanying physiotherapy with lymphatic drainage, toe mobilisation, gait training, and isometric exercises were obligatory. Orthopaedic insoles were generally fitted 6 months post-operatively.

### 2.4. Assessment Methods

The Foot Function Index-D (FFI-D) with 18 items (10 items for disability, 8 items for pain) and the SF-12 questionnaire, with the physical and mental component summaries, were collected by the treating surgeon in order to assess clinical outcomes. Demographic data including BMI, pre-existing conditions such as diabetes mellitus and arterial hypertension (metabolic syndrome-associated), and nicotine abuse were obtained for each patient. In addition, a comparison of the pre-operative and post-operative radiographs was performed. Of particular interest were the dorsoplantar (dp) talocalcaneal angle, dp talo–first metatarsal angle, talonavicular coverage angle, lateral talocalcaneal angle, and talus–first metatarsal angle on the lateral view. Radiographic measurements were taken independently by a radiologist and treating surgeon.

### 2.5. Statistical Analysis

When this study was planned, reviews of other studies focusing on a similar question revealed a comparable number of patients [10,16,17,18,19]. Statistical analyses were performed using SPSS v. 23 software (IBM Dtl. GmbH, Ehningen, Germany). The primary objective was to demonstrate the non-inferiority of FFI-D of either group at the follow-up stage after 3.4 years [20]. Furthermore, descriptive and explorative statistical analyses for the queried scores and evaluations of the pre- and post-operative radiographs (including within-group means, medians, minima and maxima, and standard deviations) were applied. Student’s *t*-test and ANOVA were used. The power of the study was 0.8, and significance level was set to *p* < 0.05, with a 95% confidence interval.

## 3. Results

The mean FFI-D was 33.9 (group A: 34.5; group B: 33.5); the mean SF-12 physical component summary was 43.13 (group A: 40.9; group B: 44.9); and the mean SF-12 mental component summary was 43.13 (group A: 40.9; group B: 44.9) (Table 2). No significant differences were observed between the groups. The post-operative results demonstrated that the angles (talometatarsal and talocalcaneal in the anteroposterior and lateral view) were largely comparable (Table 3). Only the lateral talocalcaneal angle showed a significant difference between the groups (group A: 50.5°; group B: 46.1°; *p* = 0.047), norm value < 50° [21]. All reconstructed angles were also compared with the clinical scores. No correlation could be found either in the respective group or with all study patients together (e.g., FFI-D versus post-operative dorsal talometatarsal angle, r = 0.023).

Both surgical procedures could be performed in a comparable operative time, which was 112 min on average (group A: 110; group B: 144; *p* > 0.05).

All patients reported strict adherence to the prescribed rehabilitation protocol, so no significant differences were evident between the groups. All procedures were performed by two equally experienced surgeons. The clinical and radiographic results of the two surgeons did not differ considerably; hence, a separate presentation of the results was omitted.

### Complications

In group A, intraoperative injury occurred to the artery dorsalis pedis. This was followed by microsurgical reconstruction in the same operation. In two cases, screws were repositioned along the talonavicular joint before the resumption of full weight bearing. A total of four pseudarthrosis of the talonavicular joint were identified, two cases each in groups A and B. In these patients, revision with autologous cancellous bone grafts was performed 8, 9, 11, and 14 months post-operatively. In group B, three patients complained of persistent swelling beyond 6 months.

## 4. Discussion

The most important finding of our study was that the two presented surgical options for the treatment of AAFD due to posterior tibial tendon dysfunction based on the clinical scores in the evaluated patient cohort were equivalent. The surgical procedures presented correspond to the recognised treatment of progressive collapsing foot deformity [22]. The values obtained from the FFI-D are difficult to compare with those reported in the literature, as studies with smaller sample sizes mainly included patients with minor deformities [18,23]. Ebaugh et al. reported a remarkable improvement of the FFI-D, from 52.1 to 10.3 points. The comparatively small number of patients (*n* = 16) and the inclusion of patients with mild deformities (Stage IIa) should be noted; Stage 3 deformities were excluded [24]. The present study is characterized by the fact that only patients with higher-grade deformities (Stage IIb and IIIa, b according to Raikin et al.) were included [6].

In a study by Jeng et al. [25], in which the triple arthrodesis was combined with a soft-tissue balancing procedure, the results of the SF-12 questionnaire showed similar results. The current values are somewhat lower than those obtained by Jeng et al., with respect to the mental component summary, but higher than those from the physical component summary [25]. Bolt et al. [26] reported a mean “Meary’s angle” of 11.1° in the early post-operative phase and only 18° during the latest follow-up. However, 17 patients received an isolated calcaneal displacement osteotomy with no additional talonavicular arthrodesis, as in the present study. In the authors’ view, this difference confirms both procedures presented in the study as adequate treatments. However, the authors’ experience is consistent with the prevailing opinion in the literature that talonavicular arthrodesis is the standard of surgical treatment for AAFD [27,28,29,30]. Therefore, radiological results should not be considered in isolation from the clinical outcome.

Fadle et al. [10] compared the double versus triple arthrodesis for adult-acquired flatfoot deformity. A key difference in the study by Fadle et al. is the significantly younger population (mean age in the double and triple arthrodesis groups: 25 and 20 years respectively). In addition, the angles measured pre-operatively were significantly closer to normal than in the present study.

Encouragingly, the angles were shown to be comparable to the post-operative values otherwise obtained in joint-preserving procedures in young adults [16]. The talo–first metatarsal angle on lateral view, the so-called Meary’s angle, represents one of the most important radiological criteria regarding the assessment of flatfoot deformity. Thus, in the present study, a moderate-to-severe deformity could be converted into a physiological configuration or at most a mild deformity in all patients, irrespective of the group allocation [31].

A subgroup analysis of patients with a remaining deformity smaller or larger than 10° in the talo–first metatarsal angle on lateral and dorsoplantar view did not reveal any conclusions on the clinical outcome. In both groups, two cases suffered a nonunion of the talonavicular joint. Only one out of four patients was a smoker. This confirms the higher rate of pseudarthrosis in the talonavicular joint than other foot joints [15,32,33,34]. Burrus et al. reported a nonunion rate of 44% [35]. The obvious causes are the generally critical blood supply to the talus as well as the sclerotic lesions of the os naviculare in patients with AAFD due to incorrect loading.

In the present study, all cases healed after a single revision with the insertion of autologous cancellous bone. To date, no particular osteosynthesis (screw, Nitinol compression implant, or plate) has shown superiority in terms of fusion rate at the talonavicular joint. Fortunately, nonunion at the subtalar joint was not evident. The authors’ experience is consistent with that reported in the literature; the number and diameter of the screws inserted do not influence the fusion rate [36,37,38].

For the IIb and IIIa, b deformities according to Raikin et al. presented here, combined talonavicular arthrodesis and calcaneal displacement osteotomy is a relevant alternative procedure to the established conventional or minimally invasive double arthrodesis [39,40]. The study is therefore suitable as a basis for a prospective comparison of the two treatment options presented.

### Limitations

This was a monocentric study with a retrospective design. Clinical scores were not collected pre-operatively.

Valid criteria for the classification of AAFD are still lacking. The assessment of posterior tibial tendon dysfunction, according to Johnson and Strom, is primarily based on clinical examination and available imaging (ultrasound and MRI). Histopathologic evaluation is rarely available at the time of classification. Thus, a high degree of intraobserver and interobserver variability is apparent; this situation could affect the inclusion and exclusion criteria.

The indication for surgery was also seemingly influenced by the experience and preference of the surgeon. Consequently, the decision to choose one of the comparative procedures was influenced by the surgeon’s experience.

The group of patients treated by double arthrodesis tended to be older, with a correspondingly lower activity requirement in their lifestyle. This could have influenced the scores in the collected data.

## 5. Conclusions

Based on the available data, our study confirmed that the combined procedure of talonavicular arthrodesis and calcaneal shift, with preservation of the subtalar joint, can be considered equivalent to the established double arthrodesis, with no significant differences in terms of clinical and radiological outcome. The study is therefore suitable as a basis for a prospective comparison of both procedures.

## Figures and Tables

**Figure 1 jcm-11-00840-f001:**
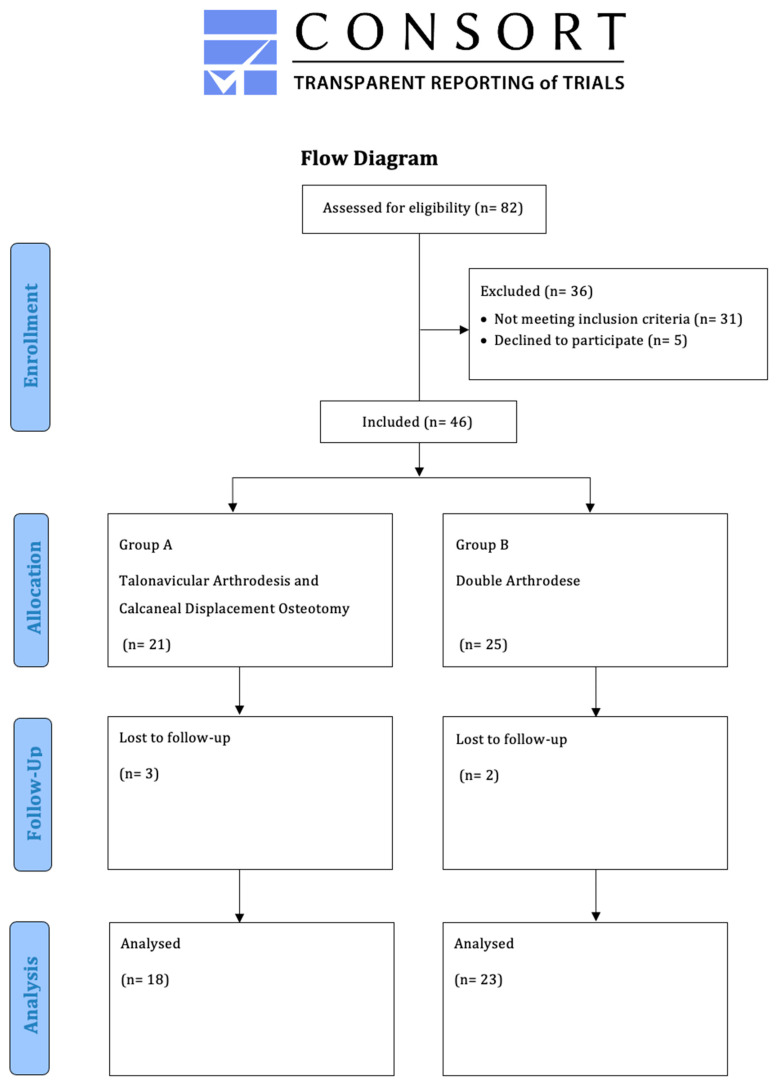
Flow chart.

**Figure 2 jcm-11-00840-f002:**
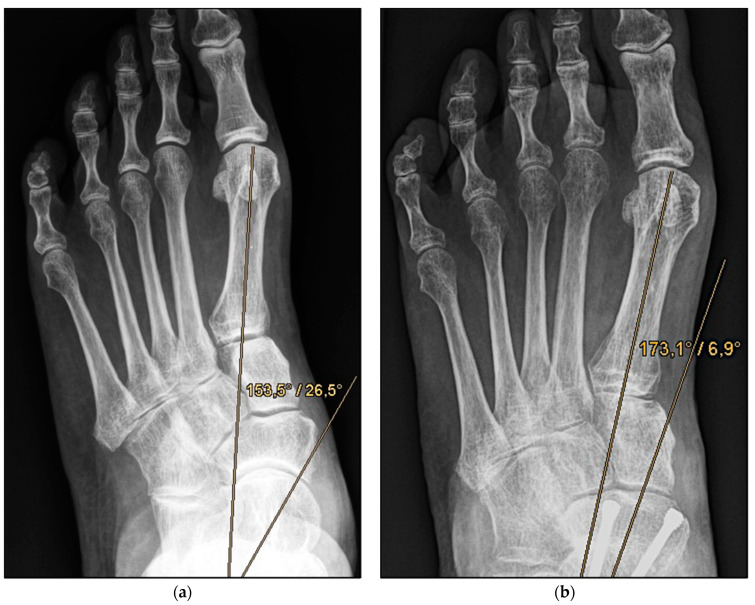
Pre- and post-operative radiographic findings of combined talonavicular arthrodesis and calcaneal displacement osteotomy, left foot. (**a**) Anteroposterior view pre-operative, (**b**) anteroposterior view post-operative.

**Figure 3 jcm-11-00840-f003:**
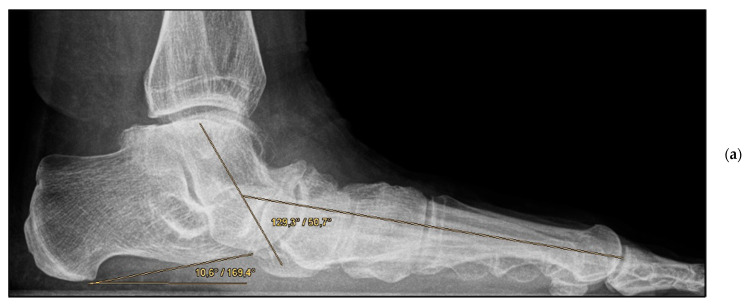
Pre- and post-operative radiographic findings of combined talonavicular arthrodesis and calcaneal displacement osteotomy, left foot. (**a**) Lateral view pre-operative, (**b**) lateral view post-operative.

**Figure 4 jcm-11-00840-f004:**
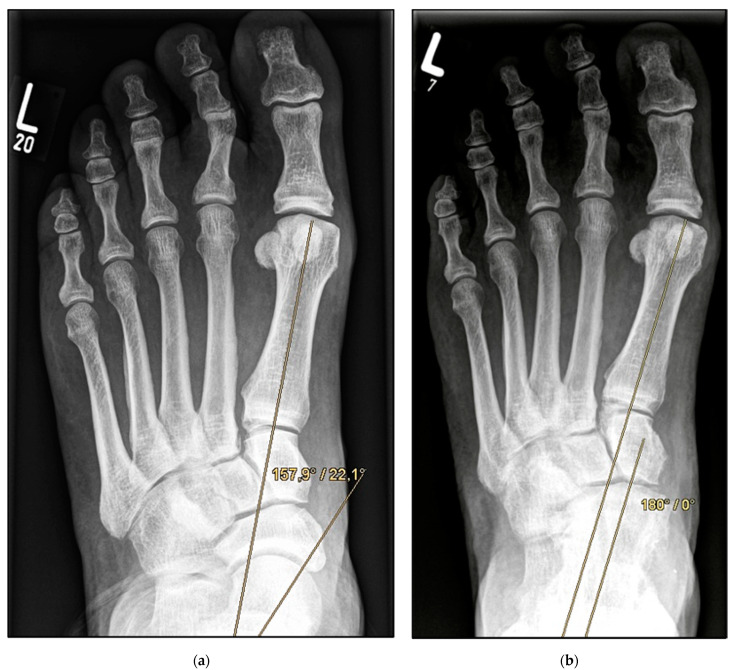
Pre- and post-operative radiographic findings of double arthrodesis, left foot. (**a**) Anteroposterior view pre-operative, (**b**) anteroposterior view post-operative.

**Figure 5 jcm-11-00840-f005:**
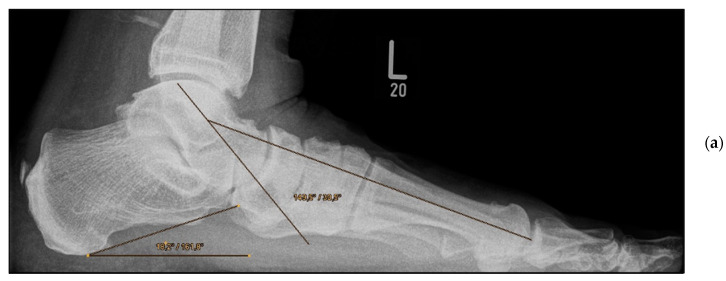
Pre- and post-operative radiographic findings of double arthrodesis, left foot. (**a**) Lateral view pre-operative, (**b**) lateral view post-operative after implant removal.

**Table 1 jcm-11-00840-t001:** Patient characteristics (with subgroups).

Characteristic	Group A (*n* = 18)	Group B (*n* = 23)	All (*n* = 41)	*p*
Age, years				
Mean	60.06	66.04	63.42	0.073
SEM	2.22	2.29	1.66	
Minimum	45.00	35.00	35.00	
Maximum	78.00	81.00	81.00	
Sex				
Male, n (%)	4 (22.2)	9 (39.1)	13 (31.7)	0.259
Female, n (%)	14 (77.8)	14 (60.9)	28 (68.3)	
BMI, kg/m^2^				
Mean	27.55	28.35	28.00	0.548
SEM	0.86	0.94	0.65	
Minimum	21.80	21.60	21.60	
Maximum	35.40	36.90	36.90	
Affected side				
Left, n (%)	12 (66.7)	15 (65.2)	27 (65.9)	0.925
Right, n (%)	6 (33.3)	8 (34.8)	14 (34.1)	
Smoker				
Yes, n (%)	10 (55.6)	5 (21.7)	15 (36.6)	0.026
No, n (%)	8 (44.4)	18 (78.3)	26 (63.4)	
Pre-existing conditions				
Metabolic syndrome-associated, n (%)	6 (33.3)	17 (73.9)	23 (56.1)	0.018
Others, n (%)	8 (44.5)	4 (17.4)	12 (29.3)	
None, n (%)	4 (22.2)	2 (8.7)	6 (14.6)	
Orthopaedic insoles, n (%)				
Only pre-operative	2 (11.1)	0 (0.00)	2 (4.9)	
Only post-operative	3 (16.7)	4 (17.4)	7 (17.1)	
Pre- and post-operative	13 (72.2)	18 (78.3)	31 (75.6)	
Never	0 (0.00)	1 (4.3)	1 (2.4)	

Group A, calcaneal shift and talonavicular arthrodesis; group B, double arthrodesis; BMI, body mass index; SEM, standard error of the mean.

**Table 2 jcm-11-00840-t002:** Radiographic angles, immediately pre- and after an average of 3.4 years post-operative.

Measurement	Group A (*n* = 18)	Group B (*n* = 23)	All (*n* = 41)	*p*
Pre-operative				
AP talocalcaneal angle in degrees				
Mean (range)	34.5 (23.0–47.0)	31.6 (17.0–45.0)	32.9 (17.0–47.0)	0.223
SEM	1.61	1.64	1.17	
AP talo–first metatarsal angle in degrees				
Mean (range)	25.8 (15.0–36.0)	25.2 (7.0–42.0)	25.5 (7.0–42.0)	0.831
SEM	1.68	2.16	1.40	
Talonavicular coverage angle in degrees				
Mean (range)	32.8 (17.0–51.0)	31.7 (20.0–50.0)	32.2 (17.0–51.0)	0.714
Minimum	2.09	1.88	1.38	
Lateral talocalcaneal angle in degrees				
Mean (range)	62.1 (45.0–70.0)	58.7 (38.0–71.0)	60.2 (38.0–71.0)	0.208
SEM	1.52	2.03	1.32	
Meary’s angle in degrees				
Mean (range)	30.4 (14.0–53.0)	29.1 (11.0–49.0)	29.7 (11.0–53.0)	0.700
SEM	2.22	2.55	1.71	

Post-operative				
AP talocalcaneal angle in degrees				
Mean (range)	23.2 (17.0–35.0)	20.6 (7.0–30.0)	21.7 (7.0–35.0)	0.180
SEM	1.32	1.36	0.97	
AP talo–first metatarsal angle in degrees				
Mean (range)	6.4 (0.0–20.0)	8.7 (0.0–22.0)	7.5 (0.0–22.0)	0.333
SEM	1.47	1.84	1.17	
Lateral talocalcaneal angle in degrees				
Mean (range)	50.5 (39.0–62.0)	46.1 (39.0–55)	48.1 (39.0–62.0)	0.047
SEM	1.77	1.24	1.10	
Meary’s angle in degrees				
Mean (range)	11.3 (0.0–19.0)	11.2 (0.0–28.0)	11.2 (0.0–28.0)	0.985
SEM	1.34	1.65	1.08	

Group A, calcaneal shift and talonavicular arthrodesis; group B, double arthrodesis; AP, anteroposterior; SEM, standard error of the mean; Talonavicular coverage angle, the angle between articular surface of the talar head and articular surface of the proximal navicular; Meary’s angle, talo–first metatarsal angle latera. All angles are based on weight-bearing radiographs.

**Table 3 jcm-11-00840-t003:** Outcome I (with subgroups).

Measurements	Group A (*n* = 18)	Group B (*n* = 23)	All (*n* = 41)	*p*
FFI-D (18 items)				
Mean	34.45	33.51	33.93	0.897
SEM	5.53	4.71	3.54	
Minimum	1.23	3.09	1.23	
Maximum	78.40	89.51	89.51	
FFI-D pain subscale (8 items)				
Mean	29.12	21.64	24.92	0.333
SEM	5.42	5.26	3.79	
Minimum	0.00	0.00	0.00	
Maximum	71.43	84.72	84.72	
FFI-D disability subscale (10 items)				
Mean	38.36	43.38	41.17	0.532
SEM	6.15	5.16	3.93	
Minimum	0.00	5.56	0.00	
Maximum	85.56	93.33	93.33	
SF-12 (Physical component summary)				
Mean	40.91	44.87	43.13	0.243
SEM	2.51	2.22	1.67	
Minimum	25.82	13.27	13.27	
Maximum	58.10	59.46	59.46	
SF-12 (Mental component summary)				
Mean	40.91	44.87	43.13	0.059
SEM	2.51	2.22	1.67	
Minimum	25.82	13.27	13.27	
Maximum	58.10	59.46	59.46	

Group A, calcaneal shift and talonavicular arthrodesis; group B, double arthrodesis; FFI-D, Foot Function Index (rating scale for each item from zero to nine); SF-12, Short Form health survey; EFAS, European Foot and Ankle Society Score; SEM, standard error of the mean.

## Data Availability

All data intended for publication are included in the manuscript.

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
