# Peer review of "Adult-Acquired Flatfoot Deformity: Combined Talonavicular Arthrodesis and Calcaneal Displacement Osteotomy versus Double Arthrodesis"

_jcm, 2022, doi:10.3390/jcm11030840_

Round 1

Reviewer 1 Report

This is an interesting study, however, the lack of validation of the different classifications for acquired flatfoot deformity in adults should be taken into account when evaluating any research study on flatfoot deformity. The lack of evidence that these systems possess a high degree of interobserver and intraobserver reliability means that readers should use them with great care. To the extent that one observer may classify a patient as having one stage, another observer may perceive the same patient as having a different stage deformity, which may result in differences in care. The same problem could arise when a reader wishes to interpret research using these systems and apply that work to their practice. This circumstance should be considered as a limitation when analyzing the results of the study, taking into account that the indication for surgery was presumably influenced by the experience and preference of the surgeon.

Author Response

Dear Reviewer,

thank you very much for the helpful hint. Indeed, this is a known problem and makes it difficult to choose the appropriate treatment.

This fact was highlighted as a limitation:

Reply

Added Line 262: “Valid criteria for the classification of AAFD are still lacking. Assessment of posterior tibial tendon dysfunction according to Johnson & Strom is primarily based on clinical examination and available imaging (ultrasound and MRI). Histopathologic evaluation is rarely available at the time of classification. Thus, there is a high degree of intraobserver and interobserver variability. This circumstance could relevantly influence the inclusion and exclusion criteria.”

The authors

Reviewer 2 Report

Manuscript Journal of Clinical Medicine, entitled "Adult-acquired Flatfoot Deformity: Combined Talonavicular 2 Arthrodesis and Calcaneal Displacement Osteotomy versus 3 Double-Arthrodesis." 

Reviewer
General comments

It is an article with regard to the comparative study of treatments in Adult-acquired Flatfoot Deformity. The article is well written and there are only a few comments as the following.

  1. Specific comments -
    1. Introduction Page 2 line 53-54, the author stated the current practice in AAFD now favor the double arthrodesis instead of triple arthrodesis, but didn’t mention the role of combined calcaneal osteotomy with talonavicular arthrodesis. The authors should add literatures review of combined procedure to support their rationales to compare these two feasible surgical methods.
  2. Patients and Methods, Page 4 line 95-96, How did authors decide which surgical procedure should be performed in patients?
  3. Patients and Methods, the article didn’t mention who measured the radiographic parameters and responsible for functional score?
  4. Results, how about the mean surgical time in each surgical methods ?
  5. Discussion, page 10, line 212-213. The authors should discuss the opinion about with or without TN arthrodesis.

Author Response

Dear Reviewer,

thank you very much for the helpful hints. Below are the proposed changes:

  1. Introduction Page 2 line 53-54, the author stated the current practice in AAFD now favor the double arthrodesis instead of triple arthrodesis, but didn’t mention the role of combined calcaneal osteotomy with talonavicular arthrodesis. The authors should add literatures review of combined procedure to support their rationales to compare these two feasible surgical methods.

Reply Added Line 60: “Biomechanical studies and clinical experience reports already confirmed the combined procedure as an adequate treatment for AAFD.”[1,2]

  1. Patients and Methods, Page 4 line 95-96, How did authors decide which surgical procedure should be performed in patients?

Reply: The decision was made by the surgeon. To maintain comparability of the population in the study, only bony procedures were included. Patients with an additional soft tissue balancing procedure were excluded. Several references were made to this in the manuscript.

Abstract

“All patients underwent isolated bony realignment of the deformity:…”

Introduction:

“In the present study, we compared two isolated bony realignment for AAFD:…”

Patients and methods

“All patients underwent isolated bony realignment of the deformity:…”

Exclusion criteria

The same applied to patients who, in addition to bony realignment, received a soft tissue balancing procedure…”

  1. Patients and Methods, the article didn’t mention who measured the radiographic parameters and responsible for functional score?

Reply Added 148: “The Foot Function Index-D (FFI-D) with 18 items (10 items for disability, eight items for pain) and the SF-12 questionnaire with the physical and mental component summary were collected by treating surgeon to assess clinical outcome.”

“Radiographic measurements were taken by an independent radiologist and treating surgeon.”

  1. Results, how about the mean surgical time in each surgical methods?”

Reply Added Line 175: “Both surgical procedures could be performed in a comparable operative time, which was 112 min on average (group A: 110, group B: 144; p>0.05).”

  1. Discussion, page 10, line 212-213. The authors should discuss the opinion about with or without TN arthrodesis.

Reply Added Line 224: “However, the authors' experience is consistent with the prevailing opinion in the literature that TN arthrodesis is one of the most effective procedures in the surgical treatment of AAFD. [3-6]Therefore, radiological results should not be considered in isolation from the clinical outcome.”

The authors

  1. Johnson, J.E.; Yu, J.R. Arthrodesis techniques in the management of stage II and III acquired adult flatfoot deformity. Instr Course Lect 2006, 55, 531-542.
  2. Malik, A.; Grant, E.; Rhodenizer, J. Analysis of Micromotion in a Talonavicular Arthrodesis With and Without a Calcaneal Displacement Osteotomy in a Cadaver Model. J Foot Ankle Surg 2020, 59, 91-94, doi:10.1053/j.jfas.2018.10.007.
  3. Catanzariti, A.R.; Adeleke, A.T. Double arthrodesis through a medial approach for end-stage adult-acquired flatfoot. Clin Podiatr Med Surg 2014, 31, 435-444, doi:10.1016/j.cpm.2014.04.001.
  4. Fornaciari, P.; Gilgen, A.; Zwicky, L.; Horn Lang, T.; Hintermann, B. Isolated talonavicular fusion with tension band for Muller-Weiss syndrome. Foot Ankle Int 2014, 35, 1316-1322, doi:10.1177/1071100714548197.
  5. Peng, Y.; Niu, W.; Wong, D.W.; Wang, Y.; Chen, T.L.; Zhang, G.; Tan, Q.; Zhang, M. Biomechanical comparison among five mid/hindfoot arthrodeses procedures in treating flatfoot using a musculoskeletal multibody driven finite element model. Comput Methods Programs Biomed 2021, 211, 106408, doi:10.1016/j.cmpb.2021.106408.
  6. Vacketta, V.G.; Jones, J.M.; Philp, F.H.; Saltrick, K.R.; McMillen, R.L.; Hentges, M.J.; Catanzariti, A.R. Radiographic Outcomes of Talonavicular Joint Arthrodesis With Varying Fixation Techniques in Stage III Adult Acquired Flatfoot Reconstruction. J Foot Ankle Surg 2021, doi:10.1053/j.jfas.2021.12.022.